# Physicochemical, Functional, and In Vitro Digestibility of Protein Isolates from Thai and Peru Sacha Inchi (*Plukenetia volubilis* L.) Oil Press-Cakes

**DOI:** 10.3390/foods11131869

**Published:** 2022-06-24

**Authors:** Saroat Rawdkuen, Stefano D’Amico, Regine Schoenlechner

**Affiliations:** 1Unit of Innovative Food Packaging and Biomaterials, School of Agro-Industry, Mae Fah Luang University, Chiang Rai 57100, Thailand; 2Institute of Animal Nutrition and Feeding, AGES—Austrian Agency for Health and Food Safety, Spargelfeldstr. 191, 1220 Vienna, Austria; stefano.d-amico@ages.at; 3Department of Food Sciences and Technology, Institute of Food Technology, BOKU-University of Natural Resources and Life Sciences, Muthgasse 18, 1190 Vienna, Austria; regine.schoenlechner@boku.ac.at

**Keywords:** Inca peanut, plant protein, techno-functional properties, simulated trypsin digestion, functional ingredient, PUFA, defatted meal

## Abstract

Proteins from Sacha inchi (SI) have been widely known for their health-benefiting properties. This study aimed to investigate the different protein isolates obtained from oil press-cakes of Thai and Peru SI. The protein content and protein recovery of Thai and Peru SI were estimated to be 93.27, 90.67%, and 49.15, 59.32%, respectively. The protein patterns of the Thai and Peru SI samples analyzed by SDS-PAGE showed glycoprotein as a major protein, with a molecular weight of 35 kDa. Both protein isolates (PI) showed water and oil holding capacities in the range of 2.97–3.09 g/g sample and 2.75–2.88 g/g sample, respectively. The emulsifying properties of the PI from Thai SI were higher than those of Peru (*p* < 0.05), while the foaming properties were not analogous to the emulsion properties. The Thai SI sample showed lower digestibility up to 120 min of in vitro digestion time than that of the Peru SI sample (*p* < 0.05). However, simulated in vitro pepsin digestion of Thai and Peru Si samples displayed hydrolyzed protein bands compared to trypsin digestion, which showed no protein patterns in both SI samples on a 4–20% gradient gel. These results suggest that the protein isolates from Thai and Peru SI exhibit marked variations in physical and techno-functional properties and have a high potential to be employed as plant-based protein additives for future non-animal-based protein-rich foods.

## 1. Introduction

Concerns about food security, as well as the challenges of a growing global population, encourage researchers to look for sustainable and environmentally friendly alternative protein sources [1]. The global protein demand is increasing, necessitating the addition of resources to those already available. Globally, animals account for 40% of humans’ protein consumption compared to 60% being consumption of plant proteins [2]. Plant proteins, comparatively, are becoming more popular as an alternative to animal proteins all over the world. In the context of today’s limited supply chain, animal-based proteins are expensive and are directly linked to climate change, freshwater depletion, biodiversity loss, greenhouse gas emissions, and health risks [3]. Since the previous decade, the food industry has been rapidly responding to widen the scope of plant-based proteins as functional meat analogues [1]. The number of plant-based meat analogues that have replaced animal-based products, in whole or in part, has surpassed 4400 products worldwide [2].

*Plukenetia volubilis*, commonly known as Sacha inchi (SI), Sacha peanut, mountain peanut, or Inca peanut, is a plant of the *Euphorbiaceous* family, which grows in the Amazon rainforest, as well as some of the Windward Islands in the Caribbean [3]. This plant has long been cultivated in Peru as a native crop. Nowadays, it is cultivated widely in the northern and north-eastern parts of Thailand and other countries [4]. SI is primarily a rich source of oil, with a high polyunsaturated fatty acid (PUFA) content, such as α-linolenic acid (53.8%), linoleic acid (33.4%), and tocopherols concentration (2540.1 mg/kg), being constituted of γ-tocopherol (64.7%) and δ-tocopherol (35.3%) [5]. SI oil press-cakes after oil extraction remain as waste by-products for valorization, which account for 56.61% of the protein content, which contains essential amino acids, including lysine, leucine, histidine, and phenylalanine [4]. SI oil press-cakes could be a potential alternative source of plant-based proteins, which have received great attention from consumers recently due to consumers’ increasing anxieties regarding food security and the rising cost of animal-derived proteins [6]. As a protein analogue, potato juice protein concentrate (PJPC) contained significant amounts of Fe, Mn, K, Cu, amino acids (lysine and threonine), and exhibited high antioxidant potential, which was followed by growth inhibition of a gastric cancer cell line (Hs 746T), a colon cancer cell line (HT-29), and human colon normal cells (CCD 841 CoN) [7]. In addition, PJPC and fiber maintained gut health in which nutrients’ digestibility and the nutritive value of the diet were not affected during the digestion process [8]. The extraction technique is an important factor for safeguarding the functionality of plant proteins. Brewer’s spent grains (BSG) were dried with a vacuum microwave (48 min) and oven drying (50 min) at 65 °C. The extracted protein with a shorter time demonstrated higher overall acceptability and moderate water and oil holding capacities [9]. BSG contain hydroxycinnamic acids, including ferulic acid, *p*-coumaric acid, and caffeic acid, which have shown bioactivity in the pure form, such as antioxidant, anti-inflammatory, anti-atherogenic, anti-cancer, and prevention of oxidant-induced DNA damage, possibly by Fe chelation [10].

Proteins are often used as food ingredients for their functional properties, such as solubility, water holding, and oil binding capacities, as well as foaming and emulsion properties, to impart desirable quality attributes in the final product [11]. The functional properties of proteins are intrinsic in nature, which affect the behavior of proteins in the food system during processing, manufacturing, and storage [12]. Some of these properties depend on the physicochemical characteristics of proteins, including molecular weight, amino acid composition, net charge, and surface hydrophobicity [13]. Plant proteins play a significant role in modifying the desired techno-functional properties during food product processing and formulation. Additionally, milk protein concentrate was blended with soy, rice, and pea protein isolates, with the soy protein isolate mixed with milk protein concentrate showing higher simulated gastric phase digestibility [14]. SI protein that included SI protein concentrate and protein hydrolysate prepared by crude papain and Calotropis proteases showed higher biological activity, including antioxidant properties, and were rich in essential amino acids [15]. SI whole seed cakes were reported to contain higher crude protein compared to tea seed press-cakes, which were highest in carbohydrates, and, meanwhile, both the press-cakes showed higher bioactivity [4]. The techno-functional properties, including the solubility, water holding capacity, oil binding capacity, emulsifying capacity, emulsifying stability, foaming capacity, and foaming stability, of the SI protein isolate obtained via alkaline and isoelectric precipitation showed higher magnitudes compared with the soy protein isolate [16]. Therefore, the objectives of this study were to prepare protein isolates from Thai and Peru SI oil press-cakes using an alkaline-acid precipitation method. Extraction efficiency was estimated in the Thai and Peru protein isolates, including protein content, extraction yield, and protein recovery. In addition, the physicochemical and functional properties and simulated sequential pepsin–trypsin digestion were also investigated. The protein structures before and after digestion of the SI protein isolates were determined by SDS-PAGE. Thus, the SI protein isolates were initially prepared from the Thai and Peru oil press-cakes and compared regarding the aforementioned properties in order to report their potential as plant-based proteins for future applications in the development of meat analogues.

## 2. Materials and Methods

### 2.1. Chemicals and Samples

Sigma Chemical Co. provided trypsin (from bovine pancreas), pepsin (from porcine gastric mucosa), dithiothreitol (DTT), and Sigma marker wide range (St. Louis, MO, USA). Fluka Chemica-Biochemika supplied the sodium dodecyl sulfate (SDS) (Buchs, Switzerland). Merck supplied tris-hydroxymethyl aminomethane and Coomassie Brilliant Blue R-250 (Darmstadt, Germany). RDTH (Carl Roth GmbH+ Co. KG, Karlsruhe, Germany) provided the albumin fraction V, Roti-Blue protein assay kit, and Roti-PAGE Gradient (4–20 percent) ready-to-use polyacrylamide gel. Thai Rubber Land and Plantation Co., Ltd. provided Thai Sacha inchi (T) press-cakes (62.62% protein, Chiang Rai, Thailand). The company Agroin-dustrias Osho S.A.C. provided Peru Sacha inchi (P) press-cakes (56.92% protein, Lima, Peru). The oil pressing residue was dried, milled, and sifted. This end-product was named Sacha inchi flour, whereas the residue after cold pressing was called press-cake. The SI flour samples were collected, air-dried at 60 °C for overnight, ground into fine powder, and kept at −20 °C until used for analyses.

### 2.2. Preparation of Protein Isolates by Alkaline-Acid Extraction Process

The alkaline extraction and acid precipitation method described by Namsoo et al. [17] was used to prepare protein isolates from Peru and Thailand Sacha inchi (SI). The powdered oil press-cake samples were mixed with distilled water in a 1:10 ratio, and then the pH was adjusted to 11.0 using 1 N NaOH. The samples were continuously stirred (at 100 rpm) and heated at 50 °C for 1 h. The suspended material from the SI samples was then filtered through filter paper. The supernatant was collected and adjusted to pH 4.5 with 1 N HCI before being centrifuged at 2600× *g* force for 15 min at ambient temperature. The precipitate SI samples were collected on a muslin cloth, washed with distilled water, and centrifuged at 2600× *g* force for 15 min. The obtained SI protein isolate samples were dried overnight in a hot air oven at 40 °C. Prior to analysis, the dried SI protein isolate samples were ground into powder and stored at 4 °C. The protein contents of the residual solvents of Peru and Thai SI samples after extraction are also determined and reported as 57.20 and 63.13%, respectively.

### 2.3. Extraction Efficiency Calculation

The extraction efficiencies (yield, recovery, and purity) of protein from SI were calculated as follow:

Gram of press-cake
Extraction yield (%) = Gram of protein isolates × 100

Protein content of press-cake × g of sample
Protein recovery (%) = Protein content of isolates × g of isolates × 100

Protein content in press-cake
Purity (fold) = Protein content in isolates

### 2.4. Determination of Physicochemical Properties

#### 2.4.1. Color Attributes

A portable Hunterlab instrument (10° standard observers, illuminant D65, Hunter Associates Laboratory: Reston, VA, USA) was used to measure the color of Thai and Peru SI protein isolates. White tile was used to standardize the chromameter. The parameters L*, a*, and b* represent lightness, redness, and yellowness, respectively. These attributes were also used to calculate E and whiteness. Original press-cake from Peru (L* = 65.66, a* = 0.17, b* = 11.65) and Thailand (L* = 51.02, a* = 3.02, b* = 14.96) were analyzed for reference color values.
ΔE = [(ΔL*)^2^ + (Δa*)^2^ +(Δb*)^2^]^1^^/2^
Whiteness = 100 − [(100 − L*)^2^ + a*^2^ + b*^2^]^1^^/2^

#### 2.4.2. Bulk Density

Protein isolates from Thai and Peru Sacha inchi were determined for bulk density according to the method of Kaur and Singh [18]. A ten mL graduated cylinder (previously tarred) was gently filled with Peru and Thai protein isolate powder, pre-sieved via sieve No. 72 (British Sieve Standards), and the bottom of the cylinder was tapped on a laboratory bench until no further diminution was observed. The weight of the sample per unit volume of the sample (g/mL) was calculated in triplicate.

### 2.5. Protein Content Determination

Protein content was calculated by estimating total nitrogen content using the Kjeldahl method, AOAC method number 984.13. [19]. The protein content of Peru and Thai SI protein isolates was determined using 6.25 as a conversion factor. The protein content in the residual solution after preparation of protein isolates was determined by the method of Bradford [20] using bovine serum albumin as a standard at the absorbance of 595 nm.

### 2.6. Electrophoresis Profile of Peru and Thai SI Protein Isolates

Sodium dodecyl sulfate polyacrylamide gel electrophoresis (SDS-PAGE) was used to determine the protein patterns of SI protein isolates and residual solutions [16]. The samples were mixed in a 1:1 ratio with sample buffer (0.5 M Tris-HCl, pH 6.8 containing 4 percent SDS, 20% glycerol, 0.03% Bromophenol Blue with/without 10% DTT) and allowed to boil for 3 min. Protein samples were loaded into Roti-PAGE Gradient (4–20%) precast gels and run at a constant current of 60 mA in an electrophoresis buffer tank filled with buffer solution using a Biostep^®^ GmbH power supply (Jahnsdorf, UK). The gels were stained overnight in a staining solution (Coomassie Blue R-250-methanol-acetic acid) with gentle shaking at 50 rpm. The gel was de-stained with de-staining solution I and II (methanol–acetic acid–water) until the background was clear and then dried.

### 2.7. Determination of Functional Properties

#### 2.7.1. Protein Solubility

SI protein isolates and residual solutions were investigated for protein patterns. The Peru and Thai SI protein isolate suspensions (0.2%, *w*/*v*) were dissolved in 0.1 M NaOH solution. Further, 0.1 M HCl or 0.1 M NaOH was used to adjust the pH of the protein isolate suspensions to the desired pH values (pH 2–11). For 15 min, the precipitate was separated by centrifugation at 4500× *g* force. The precipitates formed were resolubilized with 10 mL of 0.1 M NaOH. Bradford [20] was used to determine the amount of soluble proteins. The following percentage of protein solubility was calculated:Solubility (%) = 100 − [ (P/T) × 100%]
where P was the protein content of the precipitate at various pH levels and T was the total protein content of the samples.

#### 2.7.2. Water and Oil Holding Capacity

The water and oil holding capacities of Peru and Thai SI protein isolates were described by Kumar et al. [21]. In pre-weighed micro-centrifuge tubes, protein isolates (0.1 g) for water and oil holding capacity were added to distilled water (15 mL) and soy-bean oil, vortexed for 1 min, respectively. The sample mixtures were incubated at room temperature for 30 min before being centrifuged at 5000× *g* for 30 min. The pellets were weighed after the supernatants were carefully decanted. The water and oil holding capacities were measured as grams of oil or water bound to SI samples.

#### 2.7.3. Emulsifying Activity and Stability

The emulsifying properties of Peru and Thai SI protein isolate samples were determined using a modified form of the Kumar et al. [21] method. In brief, 10 mL of 1% (*w*/*v*) and 5% (*w*/*v*) protein isolate suspensions were prepared in 0.2 M phosphate buffer (pH 7) and mixed with 10 mL of palm oil for 2 min. The emulsions were centrifuged for 5 min at 1200× *g* force. To calculate the emulsifying activity, the volume of the emulsion layer was measured using the following equation:Emulsifying activity (%) = [V_E_/V_T_] × 100
where V_E_ denoted the volume of the emulsified layer and V_T_ denoted the total volume.

The samples were prepared as described above, then incubated for 30 min and heated at 80 °C to determine stability of emulsion. After centrifuging the samples at 1200× *g* force for 5 min, emulsion layer was recorded. The emulsion stability was calculated using the following equation:Emulsion stability (%) = 100 − [(R/Ri) × 100]
where R was the emulsified layer volume obtained after 30 min of heating at 80 °C and Ri was the initial emulsified layer volume before heat treatment.

#### 2.7.4. Foaming Capacity and Foam Stability

Foaming properties of the Peru and Thai SI protein isolate samples were determined according to the modified method of Kumar, et al. [21] with modification. The protein isolates (20 mg/mL) were vortexed in a high-speed mode for 5 min and transferred to a measuring cylinder. The foaming capacity of all the samples was calculated as follows:Foaming capacity (%) = [(V_2_ − V_1_)/V_1_] × 100
where V_1_ was the volume of protein solutions before whipping and V_2_ was the volume of protein solution after whipping.

The foam stability of the Peru and Thai SI protein isolate samples was determined by measuring the decrease in volume of foam as a function of time for 30 min as follows:

Volume before whipping
Foam stability (%) = [(Volume after 30 min standing − Volume before whipping)] × 100

### 2.8. In Vitro Digestibility

The in vitro digestibility of Peru and Thai SI protein isolate samples was determined using the Rawdkuen, Rodzi, and Pinijsuwan method [15]. Protein dispersion samples were mixed with pepsin (enzyme: protein ratio of 1:100, g/g) and gently stirred at 37 °C for 120 min. At 0–120 min, the obtained reaction mixture of samples was immersed in boiling water for 3 min to deactivate the enzyme activity. To stop pepsin digestion, the sample mixtures were neutralized with 1.0 mol/L NaOH. Trypsin digestion was accomplished by adding trypsin (enzyme: protein ratio of 1:100, g/g) to the neutralized pepsin-digested mixtures. Until 240 min, samples were collected using the same method as for pepsin digestion for protein determination. After 120 min at 37 °C, the protein dispersions were heated in boiling water for 10 min to stop the trypsin digestion. The protein concentration was determined using the Biuret method [22]. SDS-PAGE was also used to monitor the polypeptide hydrolysis of isolated proteins.

### 2.9. Statistical Analysis

The SPSS package was used for statistical analysis (SPSS 10.0 for Windows, SPSS Inc, Chicago, IL, USA). The data were subjected to an analysis of variance (ANOVA). Duncan’s multiple-range test was used for mean comparison. All chemical analyses (*n* = 3) were carried out in triplicate.

## 3. Results and Discussion

### 3.1. Extraction Efficiency of Thai and Peru SI Protein Isolates

The extraction efficiency of the protein isolates from Thai and Peru SI oil press-cakes are presented in Table 1. The total protein content of the alkaline-acid-extracted Peru SI protein isolate (93.3 ± 0.8%) was comparatively higher than that of the Thai SI (90.7 ± 0.4%) sample (*p* < 0.05). The extraction yield and protein recovery of protein isolates showed higher values in the Peru SI (7.0 ± 0.1%, 59.3 ± 2.5%) sample compared to the Thai SI (5.0 ± 0.7%, 49.2 ± 7.5%) sample, respectively (*p* < 0.05). However, the purities of the Peru and Thai SI protein isolates were in the range of 1.5–1.6 fold (*p* > 0.05). The alkaline-acid precipitation method of the Peru and Thai SI protein isolates revealed significant differences in the percent protein content, extraction yield, and protein recovery. This might be associated with the difference in the morphology and geographical distribution of the Peru and Thai SI samples. In addition, it may be due to the difference in the soluble protein proportions that can be solubilized and extracted via the alkaline-acid extraction method in each press-cake. The protein isolates (protein content > 90%) were prepared in a cost-effective way that revealed the Peru SI oil press-cakes could be a potential raw material more readily than the Thai SI oil press-cakes because the extraction yield was 28.6% higher than the Thai SI sample regardless of the protein content. Tea seed oil press-cake and Thai SI oil press-cakes were analyzed for crude protein content, in regard to which the Thai SI sample was reported to have 56.1% total crude protein [4]. The Peru SI protein fractionated by the Osborne fractionation process reported albumin, globulin, prolamin, and glutelin contents of 43.7, 27.3, 3.0, and 31.9%, respectively, as the total aqueous soluble proteins [16]. Ruiz, et al. [21] analyzed the protein content of Sacha inchi press-cakes from two different sources and found that the protein content of *P**. volubilis* cake was greater than that of *P**. huayllabambana* cake (59 and 46%, respectively). Higher protein content was reported in protein isolates from SI press-cakes compared to the soy protein isolates prepared via alkaline-acid precipitation [23]. The high protein content in the SI cake floor was compared to the soy protein balanced nitrogen level regarding consumption without the health risks evidenced in the orally fed human subjects [24].

### 3.2. Physicochemical Properties of Protein Isolates

The physicochemical properties, including bulk density and color attributes, of the protein isolates are shown in Table 2. It was noted that the bulk densities of the Peru and Thai protein isolates were in the range of 0.71–0.73 mg/mL (*p* > 0.05). This might be because of the protein that constituted the major portion, with only a small difference in the protein content and accuracy of analyzed samples in the graduated cylinder. The results were in line with the higher bulk density value (0.17–0.21 g/mL) of native black gram protein isolate and safflower protein isolate (0.27 g/mL) [25]. The difference in bulk density of protein samples depends on the intensity of attractive inter-particle forces, particle size, and the number of contact points [26]. All the color attributes (L*, a*, b*, ΔE, and whiteness) of the Peru and Thai protein isolates were significantly different (*p*
*<* 0.05). The brightness (L*) of the Thai SI protein isolate (37.1) was lower than that of the Peru SI (44.2) protein isolate (*p* < 0.05). Similarly, the lower whiteness values of 35.4 were displayed in the Thai SI sample compared to a whiteness of 43.34 in Peru SI sample (*p* < 0.05). The darker color was visualized in the Thai SI protein isolate due to a higher a* value of 3.7 compared to the Peru SI protein isolate (0.9). The yellowness (b*) of the Thai SI protein isolate (14.2) was higher than that of the Peru isolate (10.0) (*p* < 0.05). However, the total color difference (ΔE) of the Peru SI protein isolate was greater than 1.7 (*p* < 0.05), with that of the Thai SI sample displayed by the photographs in Table 1. This difference in color values might be related to the presence of different pigments in the seed kernel. The color values of the Thai and Peru SI protein isolates were correlated with the previously reported research of SI protein concentrate and protein hydrolysate prepared by crude papain and Calotropis proteases. The reports revealed that the protein concentrate and protein hydrolysate prepared were the highest in lightness values [15].

The molecular weight of the proteins from Thai and Peru SI oil press-cakes, residue after extraction of protein isolates, and protein isolates were determined by SDS-PAGE under reducing and non-reducing conditions. Figure 1 depicts the protein patterns of the Thai and Peru SI proteins. The major protein components in the Peru SI oil press-cakes (PC) had molecular weights (MW) of 30, 40, and 50 kDa under non-reducing conditions. The reducing conditions revealed that the MW of protein bands was in the 40–50 kDa range, while the intense bands at 20 and 30 kDa were clearly visible. This additional intensity of protein bands might be caused by the disulfide bonds in the structure of the Thai and Peru SI protein isolates because of the reducing agent, allowing the protein sub-units to unfold as detected in lower MW units.

The protein bands were also observed in the residual solvent of the Peru (P_R_) and Thai (T_R_) SI samples after extraction. The band intensity visually seemed to be lower than those of the Peru (P_I_) and Thai (T_I_) SI protein isolates analyzed in non-reducing and reducing conditions. The protein bands identified will be correlated with the solubility of the hydrophilic portion during the extraction process of SI protein isolates. The two glycosylated polypeptides with MWs of 32.8 and 34.8 kDa, detected in the storage albumin fraction of Inca peanut, were reported by Sathe, et al. [27]. Moreover, Sathe, Kshirsagar, and Sharma [16] noted that the soluble seed proteins are primarily composed of polypeptides, with MWs ranging from 6 to 70 kDa, with polypeptides in the 20–40 kDa range being the most abundant. Sacha inchi protein concentrate upon enzymatic hydrolysis was documented to have sulfur containing amino acid that shows the presence of disulfide bonds [15]. Ghribi, et al. [28] reported that protein subunits in chickpea protein isolate with MWs ranging between 21 and 45 kDa were observed in the presence of β-mercaptoethanol.

### 3.3. Functional Properties of Thai and Peru SI Protein Isolates

#### 3.3.1. Solubility

The solubility results of the protein isolates from the oil press-cakes of Thai and Peru Sacha inchi are present in Figure 2. The results show that the solubility of the protein isolates increased as a function of pH. At a pH of 5.0, the protein in the Thai SI showed minimum solubility, while the Peru SI protein isolate had less ability to solubilize at a pH of 4.0, being near to the isoelectric point (pH~4.5). However, both protein isolates markedly increased their solubility below or above the pH range of 4–6. The maximum solubility (100%) of these proteins was obviously found at a pH of ~8.0 for both the Thai and Peru SI samples. The maximum solubility could be due to more hydrophilic linkages of the protein molecules with lower hydrophobicity to achieve higher solubility [29]. On the other hand, the Thai SI sample has the lowest solubility value and aggregation occurs at a pH of 4.5 to 5. The greater the pH value’s distance from pI, the more positive and negative charges appeared on the surface of the protein molecule. Hydrophilic and hydration repulsion forces could overcome the hydrophobic interaction, allowing proteins to remain highly soluble [6]. According to the solubility profiles, protein isolates may be applied in food products with a wider range of pH than quinoa protein isolate [30] and chia seed protein isolate [31], which have lower solubility.

#### 3.3.2. Water and Oil Holding Properties of Thai and Peru SI Protein Isolates

The water and oil holding capacities (WHC, OHC) of the Thai and Peru SI protein isolates are listed in Table 2. The results show that the water holding capacities of these protein isolate samples ranged between 3.0 and 3.1 g/g, with no marked difference (*p* > 0.05). The water holding capacities of the Thai and Peru SI protein isolates were in agreement with the previously published studies. It was reported that the SI protein isolate showed a lower WHC, possibly due to a smaller number of polar amino acid groups compared to the soy protein isolate [23]. Similarly, a WHC of 2.9 g/g in chickpea protein was reported by Toews and Wang [32]. However, it has been reported that Indian black gram protein (*Phaseolus mungo* L.) and kidney bean protein have higher water holding capacities, which are in the range of 5.3 to 6.7 g/g [26]. Water holding capacity differences may be influenced by the protein composition and conformation state, as well as the ratio of polar to hydrophobic parts in the amino acid chains. One reason for a protein’s higher water holding capacity relies on the presence of phosphate and polar functional groups that enhance protein hydration [6].

The oil holding capacities of the Thai and Peru SI protein isolates were 2.8 g/g and 2.9 g/g, respectively (*p* > 0.05). The OHC ranges of different plant-based proteins were reported to be lower in pea protein (1.7–1.9 g/g), lentil protein (1.9–2.1 g/g), navy bean protein (2.0 g/g), and chickpea (1.4–2.1 g/g) [33]. Meanwhile, Wani, Sogi, Shivhare, and Gill [26] and Wani, et al. [33] found that the OHCs of Indian black gram protein (*Phaseolus mungo* L.) (5.5–6.3 g/g) and kidney bean protein (5.82–6.92 g/g) were relatively higher. Similarly, the results of OHC (6.3 mL/g) were reported in protein extracted from waste potato juice [34]. The OHC depends on the potential of the protein to entrap the oil due to formation of hydrophobic interactions between non-polar amino acid side chains and hydrocarbon chains of lipid [26]. The major differences in OHC values in the different plant proteins when compared to the Thai and Peru SI protein isolates might be due to the variation in the non-polar side chains of proteins.

#### 3.3.3. Emulsifying Properties of Thai and Peru SI Protein Isolates

The emulsifying activity and stability of Thai and Peru SI protein isolates are shown in Table 2. The Thai SI sample showed the highest emulsion activity of 56.3% compared to the values obtained in the Peru SI (49.5%) sample (*p* < 0.05). Meanwhile, the emulsion stability of the Thai SI protein isolate estimated at 94.2% was also relatively higher than the Peru SI (80.3%) sample (*p* < 0.05). The various plant-based proteins reported different values of emulsion capacities, such as in peanut protein (60–65%) and cashew nut protein (54–62%) [34,35]. Proteins emulsification depends on the ability of a particular protein molecule to migrate quickly to the oil–water interface to form a protective membrane through intermolecular interactions and could solubilize in the aqueous phase for interfacial film formation [36]. However, chemical and enzymatic modifications have been employed for protein unfolding with increased flexibility and exposure of a hydrophobic surface that could improve the hydrophilic–lipophilic balance for better emulsification [35,37]. In terms of emulsion stability, the Thai SI sample showed a higher stability of 94.2% than that of the Peru SI (80.3%) sample. Protein harvested from cashew nut showed a range of emulsion stability of 75–80% [38]. Electrostatic interactions of protein functional groups are primarily responsible for emulsion stabilization against coalescence/flocculation [9]. As a result, it could be suggested that the Thai SI could be a better potential plant-based emulsifier than the Peru SI protein isolate.

#### 3.3.4. Foaming Properties of Thai and Peru SI Protein Isolates

Thai and Peru SI protein isolates’ foaming capacity and stability are presented in Table 2. It can be seen that the Peru SI protein isolate had significantly higher capacity to form a foam structure (73.3%) than the protein isolate extracted from Thai SI (46.7%) (*p*
*<* 0.05). The attained results were in line with their protein solubility profiles, in which the Peru SI sample showed higher solubility than the Thai SI sample (Figure 2). A protein must be highly soluble in the aqueous phase in order to be flexible by unfolding and refolding around the air–water interface while reducing surface tension to form an interfacial membrane between the air bubbles and the aqueous phase [32]. The various plant-based proteins analyzed for foaming capacity included *T**. grandis* seed protein (13.33–24.67%) and cashew nut protein (28.65%) [38,39]. Similarly, the ability to stabilize the foam structure of the Thai and Peru SI protein isolates was in accordance with the results of the aforementioned foaming capacities. The foam stability values of the Thai and Peru SI samples were 51.1% and 68.9%, respectively. The various reports on the foaming stability of plant proteins include white cowpea protein isolate (68.30%), cashew nut protein (65.77–76.43%), and *T**. grandis* seed proteins (65.08–77.06%) [35,38,39]. However, the foaming properties and emulsifying properties of the Sacha inchi protein isolates were contrary. Yu, et al., [39], reported that foaming properties may not be analogous to the emulsification properties for proteins that readily migrate to the oil–water interface due to the attractive and repulsive dispersion interactions between protein molecules at the oil–water interface and air–water interface, respectively.

#### 3.3.5. Functional Properties of Plant-Based Proteins

The physical and chemical properties of a protein cause changes in protein behavior during storage, processing, preparation, and consumption. A protein’s functional properties are influenced by a variety of intrinsic and extrinsic factors. Intrinsic factors include protein shape, size, sequential amino acid composition, net charge distribution, structure (secondary, tertiary, and quaternary), hydrophobicity or hydrophilicity ratio, and the protein’s ability to interact with other components of the food matrix [40]. Extrinsic factors include pH, moisture, temperature, ionic strength, enzymes, chemical additives, and mechanical processing [41]. Understanding the functional properties of a protein is necessary for potential application in different food systems because these properties change with respect to different food systems and food products [42]. The major functional properties of food proteins are protein solubility (PS), water holding and oil absorption capacities (WHC, OAC), foaming capacity and stability (FC, FS), and emulsification properties.

Protein solubility is affected by the functional properties, such as foaming and emulsification processes. The balanced interactions of hydrophobic patches favor protein–protein linkages and reduce stability, and ions increase protein solubility by inducing protein–water linkages [40]. This equilibrium is influenced by the protein’s structural properties. The charge magnitude and hydrophobicity of the amino acid side chains, as well as the protein conformation, determine the solubility of the protein in a specific solvent. Both the pH and ionic strength affect protein solubility that is lowest at an isoelectric point because the net charge on amino acid functional groups favors protein aggregation, resulting in precipitation of proteins [43]. Other factors that influence protein solubility in different food systems include thermal unit operations, ionic strength, and freezing process [44,45].

WHC and OAC are the amount of water or oil that one gram of protein can retain. Hydrophilic functional groups from glycoside moieties conjugated with proteins favor higher WHC, and the amino acids hydrophobic groups exposed in the solvent increase OAC [16]. OAC is linked to food sensory quality, such as mouth feel and flavor retention, and increases the sensorial quality of meat-based foods [43]. Particle sizes and hydrophobicity of proteins and capillary interactions enhance the OAC. A protein’s lipid binding affinity is increased by the presence of non-polar amino acid side chains. The WHC/OAC of proteins depends on the amino acid side chains that undergo conformational changes [46].

The protein polypeptides unfold to orient in the solvent based on the polar amino acid groups that migrate to the aqueous interface. Polypeptides have the ability to form a continuous foaming film upon interaction with each other [47]. A protein’s foam-forming potential is measured as FC, which describes the fate of protein molecules after whipping or stirring to form a stable foam layer [42]. Interfacial stability is required for both FC and FS, which depends on non-polar groups and protein molecular mass, resulting in increased air bubble suspension retention and a decrease in coalescence rate [45]. Additionally, FC is also influenced by the attached carbohydrates with the protein molecules [48]. Extrinsic factors, such as pH, ionic strength, and temperature, also influence foaming properties. To create strong and stable foam, the interfacial film should be firm and almost impermeable, preventing the release of entrapped air. In terms of FC and FS, different legumes differ from one another. Shevkani et al. [35] observed a link between the solubility and foaming ability of plant-based proteins based on the polar and non-polar functional groups on the polypeptide chains.

Plant proteins are known to be potential surfactants due to their diffusion of oil/water interfacial surface, and they reorient and develop an emulsion layer holding both hydrophobic and hydrophilic moieties together [40]. Proteins can also improve emulsion stability by increasing the continuous phase viscosity and slowing the rate at which oil drops move through the emulsion [42,49]. Emulsion activity and stability depend on the soluble and insoluble protein constituents, such as fats and starch, due to differences in the polar groups. A protein’s emulsification property is determined by the abundance of lipophobic functional groups on the polypeptide chains [35]. The amount of oil required to prepare one unit of emulsion is denoted by EAI, while the resistance to structural changes during storage time is denoted by ESI [50].

### 3.4. In Vitro Digestibility of Peru and Thai SI Protein Isolates

The in vitro digestibility of protein isolate samples was investigated by a sequential pepsin–trypsin digestion process, as shown in Figure 3. The simulated pepsin digestion was able to release nitrogen, as evidenced by the increased digestibility of Peru SI protein isolate up to 21.62% between 0 and 10 min of digestion time. Meanwhile, the Thai SI protein isolate demonstrated 15.66% digestibility between 0 and 10 min of digestion time. The pepsin digestion process proceeded for 120 min to attain a digestibility of 42.64% in the Peru SI sample. Similarly, the Thai SI sample demonstrated 34.99% digestibility at 120 min of digestion time. Thereafter, digestion by trypsin enhanced the protein digestibility, estimated to be 59.65% and 52.41 in the Peru and Thai SI samples at 120 min, respectively. The digestibility of the Peru and Thai SI protein isolates was in the range of 55–60% after 120 min of digestion time. The results show that the digestibility of these Thai SI protein isolates was relatively lower than the Peru SI at 10–120 min of digestion time during the simulated pepsin–trypsin digestion process (*p* < 0.05). A similar trend was observed in the percent digestibility of the Thai and Peru SI protein isolate samples at 120 min of digestion time. The digestibility of SI protein concentrate by trypsin was reported to increase at 121–130 min of digestion, and it continued to show incremental increases greater than 50% during the 120 to 240 min period of digesting time [15]. Chirinos, et al. [51] reported that sequential hydrolysis of SI with alcalase–neutrase enzymes did not affect the bioactivity of hydrolyzed SI in terms of antioxidant and antihypertensive properties.

The contribution of protein hydrolysis to the release of soluble nitrogen during the pepsin–trypsin digestion process was investigated by SDS-PAGE, the results of which are displayed in Figure 4. It was clearly observed that SI protein isolate samples with approximately 25 and 40 kDa of MW can be detected at the beginning of the digestion (0 min), and then they were further hydrolyzed by pepsin (120 min), by which the small fragment of proteins at 15 kDa was produced. The protein with MWs of around 25 and 40 kDa may contain hydrophobic and aromatic amino acids, such as phenylalanine, tryptophan, and tyrosine, in their sequences, which is highly susceptible to peptide bonds’ cleaving ability of pepsin. Meanwhile, protein bands were not detected in polyacrylamide gel during trypsin digestion (121–180 min). This is because pepsin only digests macro-proteins into polypeptides that are further digested into smaller peptide fragments by trypsin, which are peptide molecules that are too small to be detected in 4–20% gradient gel.

Yu, Zeng, Qin, He, and Chen [39] reported that *T**. grandis* seed proteins showed digestibility in the range of 58.05–59.05%. However, raw and yellow field peas protein and quinoa protein isolates have higher protein digestibility than Sacha inchi protein at about 23.4–35.1% [30,52]. The globular structure and conformation of protein may have a significant impact on protein digestibility; additionally, the presence of anti-nutritional factors may have an impact, such as tannins and trypsin inhibitors being two anti-nutrients that are closely related to protein digestion. Tannins reduce protein digestibility by binding to proteins via hydrogen bonding and hydrophobic reactions, whereas trypsin inhibitors have a negative impact due to the irreversible binding of trypsin inhibitors to the endopeptidase trypsin to form an inactive protein complex, resulting in a decrease in protein digestibility [52]. It was reported that Sacha inchi seed contained major proteins, such as 11S globulins resistant to pepsin and pancreatin hydrolysis, and exhibited high anti-inflammatory activity under in vitro conditions [53].

## 4. Conclusions

Thai and Peru protein SI isolates were prepared by an alkaline-acid precipitation process. The protein content, extraction yield, and protein recovery attained in the Peru SI sample were 2.8, 28.6, and 17.1 times higher than those of the Thai SI sample, respectively. The L*, ΔE, and whiteness values were higher in the Peru SI sample compared to the Thai SI sample, while the lowest a* and b* values were lower than those of the Thai SI sample. There was no difference evidenced in the bulk density of the Thai and Peru SI samples. The SDS-PAGE results displayed the protein bands of the Thai and Peru SI protein isolates under reducing and non-reducing conditions, which were spotted at 35 kDa following the MW standard. The Peru SI protein isolate showed higher solubility at a pH of 4.5–5 in comparison with the Thai SI sample, which showed an aggregation of proteins followed by precipitation. The WHC and OHC of the Thai and Peru SI samples showed no significant difference. However, the emulsion capacity and stability of the Thai SI sample were 12.1 and 14.8 times higher than those of the Peru SI sample. Additionally, the foaming capacity and stability of the Peru SI sample were 36.3 and 25.8 times higher than those of the Thai SI sample. The pepsin–trypsin sequential digestion of the Thai SI protein isolate showed lower digestibility compared to the Peru SI sample. Moreover, the hydrolyzed protein bands were visible in pepsin digestion, while no bands were obtained in trypsin digestion in both the protein isolate samples screened after the in vitro digestion process. Based on the physicochemical, functional, and in vitro digestibility results, SI protein isolates can be used for future food application, especially in the development of plant-based meat analogue foods. Furthermore, SI protein isolate polypeptides will be analyzed for peptide sequencing and nutraceutical parameters in future research.

## Figures and Tables

**Figure 1 foods-11-01869-f001:**
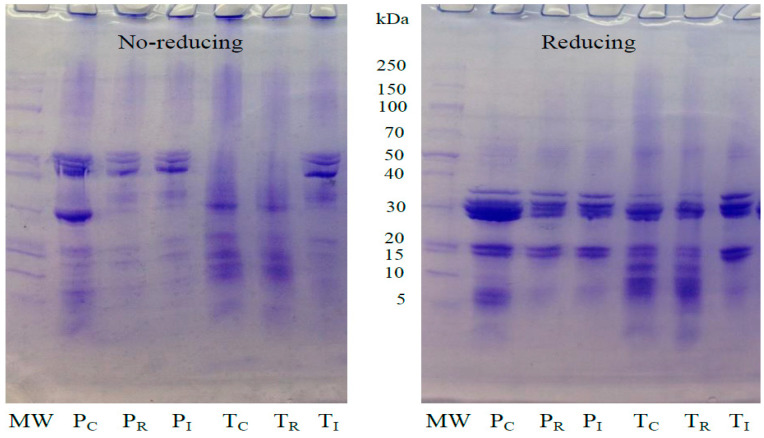
Electrophoresis patterns of protein isolates from oil press-cakes of Thai and Peru Sacha inchi samples under 4–20% gradient gel. MW: molecular weight standards; T and P: press-cake from Thai and Peru Sacha inchi, respectively. Alphabet under script: C: press-cake; R: residual after extraction; I: protein isolates.

**Figure 2 foods-11-01869-f002:**
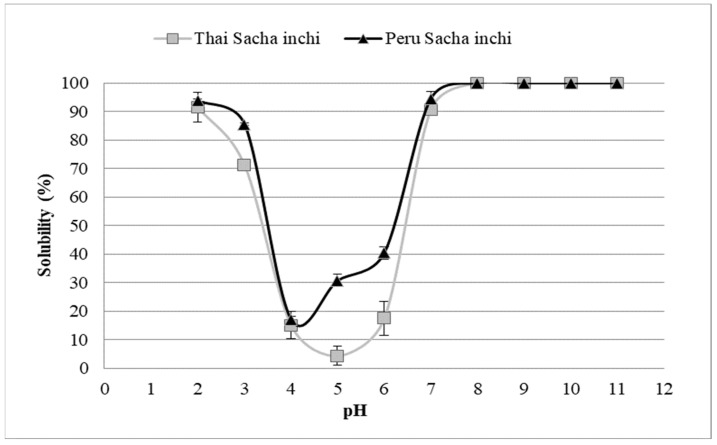
Solubility profiles of protein isolates from oil press-cake of Thailand and Peru Sacha inchi seeds.

**Figure 3 foods-11-01869-f003:**
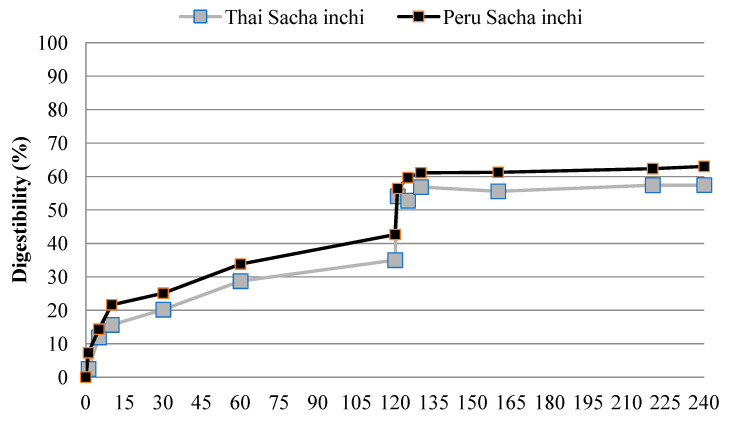
Thai and Peru protein isolates from oil press-cakes during in vitro pepsin and subsequent trypsin digestion.

**Figure 4 foods-11-01869-f004:**
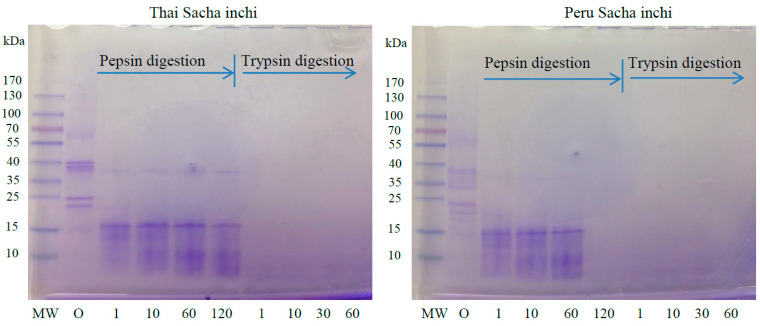
Protein patterns of sequential in vitro digestibility of protein isolates from oil press-cake of Thai and Peru SI at different times under 4–20% gradient gel. MW: molecular weight standards; numbers: incubation time at 37 °C in min.

**Table 1 foods-11-01869-t001:** Extraction efficiency of protein isolates from Thai and Peru SI oil press-cakes.

Parameters	Thai Sacha Inchi	Peru Sacha Inchi
Protein content (%, N × 6.25)	90.7 ± 0.4 ^b^	93.3 ± 0.8 ^a^
Extraction yield (%)	5.0 ± 0.7 ^b^	7.0 ± 0.1 ^a^
Protein recovery (%)	49.2 ± 7.5 ^b^	59.3 ± 2.5 ^a^
Purity (fold)	1.6 ± 0.03 ^a^	1.5 ± 0.1 ^a^

Values are mean ± standard deviation (*n* = 3). Different superscripts within the same row followed by different letters ^(a^^,b^^)^ indicate a significant difference (*p <* 0.05).

**Table 2 foods-11-01869-t002:** Comparison of physicochemical and functional properties of protein isolates from Thai and Peru Sacha inchi oil press-cakes.

Physicochemical Properties	Thai Sacha Inchi	Peru Sacha Inchi
Sacha inchi oil press-cake protein isolate	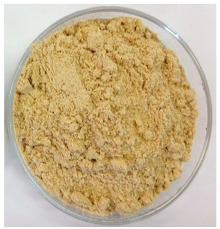	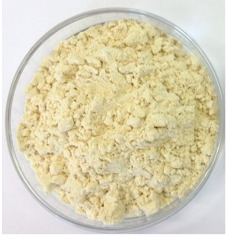
Bulk density (g/mL)	0.71 ± 0.02 ^a^	0.73 ± 0.1 ^a^
L*	37.1 ± 1.5 ^b^	44.2 ± 2.8 ^a^
a*	3.7 ± 0.1 ^a^	0.9 ± 0.1 ^b^
b*	14.2 ± 0.2 ^a^	10.0 ± 0.6 ^b^
ΔE	0.7 ± 0.02 ^b^	1.7 ± 0.6 ^a^
Whiteness	35.4 ± 1.2 ^b^	43.3 ± 2.2 ^a^
Water holding capacity (g/g)	3.1 ± 0.1 ^a^	3.0 ± 0.1 ^a^
Oil holding capacity (g/g)	2.8 ± 0.1 ^a^	2.9 ± 0.2 ^a^
Emulsifying activity (%)	56.3 ± 5.9 ^a^	49.5 ± 1.4 ^b^
Emulsion stability (%)	94.2 ± 3.1 ^a^	80.3 ± 3.9 ^b^
Foaming capacity (%)	46.7 ± 4.6 ^b^	73.3 ± 6.7 ^a^
Foaming stability (%)	51.1 ± 3.9 ^b^	68.9 ± 3.9 ^a^

Values are mean ± standard deviation (*n* = 3). Different superscripts within the same row followed by different letters ^(a^^,b^^)^ indicate a significant difference (*p <* 0.05).

## Data Availability

Not applicable.

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
