# Peer review of "Physicochemical, Functional, and In Vitro Digestibility of Protein Isolates from Thai and Peru Sacha Inchi (*Plukenetia volubilis* L.) Oil Press-Cakes"

_foods, 2022, doi:10.3390/foods11131869_

Round 1
Reviewer 1 Report
General comments:
In this study, Thai and Peru Sacha inchi protein isolates were extracted by Alkaline-acid Extraction Process, and the protein pattern, physicochemical properties, functional properties and in vitro digestion were also analyzed and determined. The reliable test results have certain reference value, the article content is rich and interesting, but the article still needs to correct some problems.
Specific comments:
1. What is the protein content of Peru and Thai SI oil-pressed cake, respectively? Please supply this data in the manuscript.
2. line 185, page 5: “Water-and oil-holding capacity” should be revised to “Water and oil holding capacity”.
3. The extraction yields of Thai and Peru SI protein isolates are very low, which is limit for their application. How to improve the extraction yield?
4. The protein bands were also observed in residual solvent of Peru and Thai SI samples after extraction. Please supply the data about the protein content of the residual solvent of Peru and Thai SI samples after extraction.
5. The mauscript revealed that “The maximum solubility could be as result of hydrophobic interactions between protein molecules are much larger than hydrophilic and hydration repulsive force produced by charged residues”. Please provide the reference about the sentence.
6. “Water and oil holding capacities (WHC) of Thai and Peru SI protein isolates” at line 359, Page 9, Please correct to “Water and oil holding capacities(WHC、OHC)of Thai and Peru SI protein isolates”.
7. Different intrinsic and extrinsic factors affect the functional properties of a protein. The authours should add some experiments, such as protein size, net charge distribution, structure, and so on, in order to better understand the functional proties of proteins.
Author Response
Please see the file attached

Reviewer 2 Report
The article "Physicochemical, Functional and In Vitro Digestibly of Protein Isolates from Thai and Peru Sacha Inchi (Plukenetia volubilis L.) Oil-Pressed Cakes" presents a method of obtaining protein isolates from oil-pressed cakes. This is an example of the use of post-production waste generated in the production of oil, which is in line with the zero waste assumptions. The paper presents selected functional and physicochemical properties in an interesting way. Nevertheless, I am asking for some clarification. Table 2 provides information on Bulk density. In the text from verse 285 to 293 the physicochemical properties are described. But I have a method related question. Lines 114 to 116 report grinding and sieving the dried residue. What sieve or set of sieves was used for the sieving and what was the maximum / minimum granulation after sieving. This has a significant impact on the value of bulk density, and unfortunately this value is missing here. Interesting article, interesting research results. I am waiting for the next article with further characteristics of proteins obtained from oil-pressed cakes Plukenetia volubilis L.
Author Response
Please see the file attached

Reviewer 3 Report
Foods
Article
Physicochemical, Functional and In Vitro Digestibly of Protein Isolates from Thai and
Peru Sacha Inchi (Plukenetia volubilis L.) Oil-Pressed Cakes
Authtrs have studied the proteins from Sacha inchi (SI) had been widely known for their heath benefiting properties.
Thai and Peru protein SI isolates were prepared by alkaline-acid precipitation process.
Protein content, extraction yield and protein recovery attained in Peru SI sample was
2.8, 28.6 and 17.1 times higher than the Thai SI sample, respectively.
Also found that protein isolates from Thai and Peru SI exhibit marked variations in physical and techno-functional properties
and has a high potential to be employed as a plant-based protein additive for future non-animal-based protein-rich foods.
An adequate number of figures and tables were given.
Structured and articulated very well.
Results were discussed properly with data support.
May have given a comparative table for such proteins or maybe with other techniques, to better it.
A literature survey must be done.
With Regards,
Author Response
Please see the file attached

Round 2
Reviewer 1 Report
This paper makes a quite comprehensive summary about physicochemical, functional and in vitro digestibly of protein isolates from Thai and Peru Sacha Inchi. Therefore, it is worth publishing and I believe that readers could draw meaningful insight from this paper.
Author Response
Thank you so much for your suggestion.
Please see the file attached for further consideration
